# The Mechanism of Action of L-Tyrosine Derivatives against Chikungunya Virus Infection In Vitro Depends on Structural Changes

**DOI:** 10.3390/ijms25147972

**Published:** 2024-07-21

**Authors:** Vanessa Loaiza-Cano, Estiven Hernández-Mira, Manuel Pastrana-Restrepo, Elkin Galeano, Daniel Pardo-Rodriguez, Marlen Martinez-Gutierrez

**Affiliations:** 1Grupo de Investigación en Ciencias Animales-GRICA, Facultad de Medicina Veterinaria y Zootecnia, Universidad Cooperativa de Colombia, Bucaramanga 680002, Colombia; vanessa.loaiza@udea.edu.co (V.L.-C.); estiven.hernandezm@udea.edu.co (E.H.-M.); 2Grupo de Investigación en Productos Naturales Marinos, Universidad de Antioquia, Medellin 050010, Colombia; mhpr2017@gmail.com (M.P.-R.); elkin.galeano@udea.edu.co (E.G.); 3Metabolomics Core Facility—MetCore, Vice-Presidency for Research, Universidad de los Andes, Bogota 111711, Colombia; d.pardorodriguez@uniandes.edu.co; 4Grupo de Investigación en Microbiología Básica y Aplicada (MICROBA), Escuela de Microbiología, Universidad de Antioquia, Medellin 050010, Colombia

**Keywords:** antiviral, chikungunya virus, in vitro, tyrosine, computational biology, mechanism of action

## Abstract

Although the disease caused by chikungunya virus (CHIKV) is of great interest to public health organizations around the world, there are still no authorized antivirals for its treatment. Previously, dihalogenated anti-CHIKV compounds derived from L-tyrosine (dH-Y) were identified as being effective against in vitro infection by this virus, so the objective of this study was to determine the mechanisms of its antiviral action. Six dH-Y compounds (C1 to C6) dihalogenated with bromine or chlorine and modified in their amino groups were evaluated by different in vitro antiviral strategies and in silico tools. When the cells were exposed before infection, all compounds decreased the expression of viral proteins; only C4, C5 and C6 inhibited the genome; and C1, C2 and C3 inhibited infectious viral particles (IVPs). Furthermore, C1 and C3 reduce adhesion, while C2 and C3 reduce internalization, which could be related to the in silico interaction with the fusion peptide of the E1 viral protein. Only C3, C4, C5 and C6 inhibited IVPs when the cells were exposed after infection, and their effect occurred in late stages after viral translation and replication, such as assembly, and not during budding. In summary, the structural changes of these compounds determine their mechanism of action. Additionally, C3 was the only compound that inhibited CHIKV infection at different stages of the replicative cycle, making it a compound of interest for conversion as a potential drug.

## 1. Introduction

Chikungunya virus (CHIKV) is the etiological agent that causes the infection known as Chikungunya fever. CHIKV is a mosquito-borne arbovirus of the genus *Aedes* spp., mainly *Aedes albopictus* and *Aedes aegypti* [1,2,3], which is widely distributed in tropical and subtropical areas below 1800 m above sea level [4]; recently, has been detected even at 2300 m above sea level (masl) [5]. This has caused the virus to spread around the world, generating outbreaks in Africa, Asia, Europe and America [6,7,8,9]. CHIKV belongs to the genus Alphavirus, of the *Togaviridae* family. It is an enveloped, spherical virus with a positive sense single-stranded RNA of approximately 12 kb, with two open reading frames (ORFs). The first ORF produces the polyprotein P1-2-3-4, which, when cleaved, gives rise to the four viral proteins necessary for the replication of genomic and subgenomic RNA (sgRNA). From the other ORF, five structural proteins are translated from the sgRNA: capsid (C), 6K, and glycoproteins E1, E2 and E3, which are cleaved by viral and cellular proteases [10].

Various types of cells have been reported to be more susceptible to infection, including epithelial and endothelial cells, primary fibroblasts, and monocyte-derived macrophages. After the inoculation of the virus (through the bite of the mosquito), there is a first round of replication, allowing its dispersion to the lymph nodes and other tissues through the circulatory system [11]. Once the infection is established, a series of nonspecific signs and symptoms occur (acute fever, headache, nausea, vomiting, myalgia, arthralgia and rash), leading in some cases to more serious symptoms (ophthalmic, neurological and occasionally cardiac), including debilitating polyarthralgia characteristics that can compromise the quality of life of patients for weeks to years, and even until death [12].

Although infections caused by this virus have been described since the 1940s in Africa, the epidemics generated in the last two decades have been the most relevant in international public health. Despite the significant reduction in the number of cases reported in tropical countries on the American continent after its emergence [13], in the last year, a new alert has been generated due to the increase in cases in the Southern Cone [14].

Therefore, due to the prevalence of the virus, the periodic nature of the epidemic, and the medium- and long-term complications of the patients, an attempt has been made to control the spread of the virus and prevent and treat the infection, as well as its effects. Thus, multiple campaigns and vector control studies have been carried out that have been insufficient [15]. Furthermore, since 2011, clinical studies have been carried out on virus-like particles (VLPs), messenger RNA (mRNA), and live attenuated vaccines, among others [16]. However, in 2023, the FDA approved the first CHIKV vaccine, Ixchiq^®^ [17], which, even in endemic areas, is not part of the vaccination program. Finally, no antiviral agent against CHIKV has been authorized for patients [18,19], and only two drugs previously licensed for other pathologies (chloroquine and ivermectin) have been proven in clinical studies. In the first case, chloroquine, despite having an in vitro effect, did not reduce the viral load in patients [20]; and, in the second case, the results have not yet been published [21].

An important source of potential new antivirals is biodiversity [22,23]. Tropical countries, including Colombia, are highly rich in their biodiversity and have been widely studied for obtaining bioactive metabolites mainly from plants [24,25], but also from Caribbean sea sponges [26]. The latter are a source of bromotyrosine-type compounds that have demonstrated antiviral activity against Dengue virus (DENV) and against Human immunodeficiency virus I (HIV-1) [27,28]. Due to the limited availability of these sponges to purify large amounts of bioactive metabolites, and their unique but simple structure, the synthesis route is the best option. This allows not only for larger amounts of compounds to be obtained, but also for different functional groups of L-tyrosine to be substituted, generating structural changes (halogens in the ring, phenolic hydroxyl groups, the esterification of the carboxyl group or the methylation of the amino group) [29]. These substitutions, in addition to the known aromatic π-electron-donating ability of tyrosine [30], could generate changes and improvements in both activity and mechanism. These synthesized compounds have shown activity against HIV-1 [31], Zika virus (ZIKV) and CHIKV [32]. In a recent study, both dibromated and dichlorinated L-tyrosine-derived phenolic compounds significantly inhibited CHIKV infection in vitro, but the mechanism of action of most of these compounds was not investigated. For this reason, in the present work, we evaluated the possible mechanism of action of six dihalogenated compounds derived from L-tyrosine, which were previously reported as antivirals against CHIKV.

## 2. Results

### 2.1. Prediction of the Pharmacokinetic Parameters of the Evaluated Compounds

The physicochemical and pharmacokinetic properties (ADME, an acronym for absorption, distribution, metabolism and excretion) of the six study compounds were predicted according to the SMILES obtained from the modeling of their chemical structures. This approach was used to determine their potential as possible drugs using the SwissADME platform (Table 1). The topological polar surface area (TPSA), calculated for C1 and C2 (Group I or primary amines), was 83.55 Å2; for C3 and C4 (Group II or tertiary amines), it was 60.77 Å2; and for C5 and C6 (Group III or quaternary amines), it was 57.53 Å2. On the other hand, the lipophilicity of the compounds was given as log P values. For C1 and C2, log P values of 0.74 and 0.91, respectively, were obtained; for C3 and C4, the log P values were 1.84 and 1.96, respectively, which were the highest values in this criterion; and for C5 and C6, the values were 0.37 and 0.58, respectively (the lowest). Similarly, when predicting lipophilicity, the solubility of the compounds was also predicted (log S (Esol)), for which values between −1.17 and −4.01 were obtained—that is, from very soluble, in the case of C1, C2 and C3, to moderately soluble, in the case of C6. Within the medicinal chemistry parameters, the synthesis accessibility (SA) values were calculated, ranging from 1.96 (C1) to 2.44 (C6); in addition, zero violations of the Lipinski rules (VLRs) were identified, as well as zero PAINS (Pan-Assay INterference CompoundS) alerts. In the case of the Brenk alerts, they were only predicted in C5 and C6.

Furthermore, according to the ADME parameters, it was predicted that all these compounds would have a high gastrointestinal absorption (GIA) and that the compounds in Group I would not cross the blood–brain barrier (BBB), while the compounds in Groups II and III could. On the other hand, in the cytochrome P450 (CYP) model, none of these compounds were inhibitors of the simulated isoforms on the platform (CYP1A2, CYP2C19, CYP2C9 and CYP3A4). Finally, as an excretion parameter, only Group III compounds could be substrates of P-glycoprotein (P-gps).

On the other hand, it is important to highlight, that based on the results of cell viability in a previous study, a concentration of 250 µM did not decrease the viability of the treated cultures by more than 10%, so this concentration was used for all subsequent antiviral evaluations.

### 2.2. Evaluation of the Antiviral Mechanism by In Vitro Strategies

As the first in vitro strategy to define the possible mechanism of action of the compounds, a direct treatment of the virus with each of the compounds was carried out. With this strategy, none of the compounds significantly inhibited the amount of IVP. Therefore, it could be concluded that none of the compounds had virucidal activity (Figure 1).

Considering that the translation of proteins and the replication of the viral genome are fundamental processes for completing the viral replication cycle, leading to the release of IVP, these three viral components (protein, genome and IVP) were evaluated in the pre-treatment and post-treatment strategies.

In the pre-treatment strategy, a statistically significant decrease in viral protein was observed in all compounds with respect to CWTs (Figure 2A). However, by quantifying the viral genome, only C4 (the brominated compound from Group II), C5 and C6 (both from Group III) significantly reduced the number of genomic copies/mL (Figure 2B). On the other hand, treatment with both compounds of Group I (C1 and C2) and with C3 of Group 3 led to a significant inhibition of IVP, while C4 and Group III compounds (C5 and C6) did not show any inhibition of this parameter (Figure 2C).

Based on these results, the ability to act in the early stages of infection was evaluated by adhesion and internalization inhibition tests, with only the three compounds that were able to reduce the release of IVP in this strategy (C1, C2 and C3). In the first trial, C1 and C3 significantly decreased the IVP with respect to the CWT, while in the second test, only C2 and C3 significantly reduced the infection (Figure 3).

In the post-treatment strategy, none of the six compounds inhibited the protein (Figure 4A) or the viral genome (Figure 4B); however, Group II (C3 and C4) and III (C5 and C6) compounds significantly decreased IVP production (Figure 4C).

For this reason, it was proposed to identify the possible stages in which these compounds act after the entry of the virus into the cells. To do this, a 24 h CHIKV/Col replication curve was obtained first, followed by a one-cycle compound addition time test according to the results of the curve. The results of the curve showed that the first cycle of viral replication occurred between 0 h and 10.5 h and the second cycle occurred between 10.5 h and 24 h, and from the end of the first replication cycle until the second cycle, the increase in IVPs was less than one logarithm (Appendix A), which is why the addition time test was only performed at the times defined for the first replication cycle (addition of compounds from 0 h to 9 h, with harvest at 10.5 h). The results showed that the four selected compounds always inhibited the number of IVPs (intra- and extracellular) at which they were added (Figure 5).

### 2.3. Evaluation of the Antiviral Mechanism by In Silico Strategies

Considering the crucial role of the fusion loop (E1 protein) in the process of CHIKV IVP internalization, the molecular docking of the two compounds that inhibited this stage (C2 and C3) with the domain that contains this loop in the viral protein (PDB: 3N42) was carried out. For this purpose, the quality of the 3N42 protein was initially evaluated using a Ramachandran plot, which revealed that 94.22% of the protein residues were found in favored regions, suggesting that the protein was of adequate quality (Appendix A) [33]. After the molecular docking of C2 and C3 with the 3N42 protein, favorable free binding energies were observed in both cases, and the affinity for C3 was greater (−6.10 ± 0.00 kcal/mol) than that for C2 (−5.83 ± 0.06 kcal/mol). Regarding the interactions necessary for the formation of the complex, it was observed that ionic and hydrogen bonding interactions play a major role in the complex (Figure 6). The structures of C2 and C3, which contain dibromophenol and dichlorophenol groups, respectively, interact with the imidazole group of HIS^226^ to establish a hydrogen bond. This type of interaction also occurs between the acidic groups of the derivatives and CYS^62^ (Figure 6C,D). In terms of ionic interactions, both molecules share the link produced by the negative charge of the acidic group and the positive charge of the amine in LYS^61^. C3 additionally establishes a second interaction between its positive charge, which is centered on the amide group, and the acidic group of GLU^247^.

Moreover, after docking studies, molecular dynamics (MD) simulations of 100 ns were performed to characterize the stability over time of the interaction between these two compounds (C2 and C3) and the 3N42 protein. In an initial assessment, the topology of the complexes was verified for C2 and C3 (Appendix A, respectively) at time points 0 ns (in cyan) and 100 ns (in red). It was observed that both compounds remained within the protein pocket at the active site of 3N42. The system stability was evaluated using the means of the RMSD (root mean square deviation) value for the protein backbone C-α in complex with L-tyrosine derivatives. According to the RMSD of each system, C3 was the most stable during the 100 ns of MD. This supports that the systems remained stable during the simulation time (Figure 6E). In contrast, for the C2 complex system, the RMSD values gradually increased after 100 ns of simulation, indicating that the C2 complex is less stable than the complex formed with C3. The RMSF (root mean square fluctuation) profiles of the complexes formed between 3N42 and L-tyrosine revealed that the interactions with compounds exhibited low mobility in most of the residues, except for the regions surrounding the active site that comprised residues (chain B, residues 170 to 225), in which the highest RMSF values were observed (Figure 6G; the residues with RMSFs greater than 0.3 or 0.4 nm are denoted in light green or red, respectively). In accordance with what was observed in the RMSD analysis, the analysis of this region revealed a greater fluctuation in the adjacent residues in the complex formed by C2 than in that formed by C3, suggesting a greater stability between 3N42 and C3. According to the analysis of hydrogen bonds during the 100 ns of simulation, complexes C2–3N42 and C3–3N42 were characterized by traversing dynamic time with the formation of at least two (2) hydrogen bonds, demonstrating the involvement of hydrogen bonding in complex formation.

## 3. Discussion

CHIKV infection has constituted one of the main public health problems in recent decades, more than just due to the symptoms it generates in its acute phase, given the complications that compromise the quality of life of patients in the medium and long term [34]. This situation has led to a deeper investigation of both the viral agent and its replicative cycle, to identify targets that can be reached by molecules with possible antiviral activity. To date, there are no approved antivirals against this infection [18]. In this sense, this study evaluated the possible anti-CHIKV mechanism of six synthetic dihalogenated compounds derived from L-tyrosine, which were selected not only for their in vitro anti-CHIKV activity but also for their previously reported in vitro and in silico safety and toxicity profiles [31,32].

The calculated values of the TPSA as a descriptor of the properties of the compounds to be transported by membranes and biological barriers within each group were the same for all the compounds, which means that there was no influence on the halogen but rather on the substitution pattern of the amino group in this parameter. Moreover, it was shown that the charged compounds are the least fat-soluble, as expected. In relation to the parameters of medicinal chemistry, the results indicate the ease of synthesis of all the compounds. In addition, no VLR or PAINS alerts were detected, suggesting that all the compounds are suitable for oral dosing and do not contain promiscuous chemical groups within their structure. According to the predictions, the least promising compounds would be those belonging to Group III as Brenk alarms, due to the charged amino group (the presence of quaternary nitrogen, as it is highly reactive) and predicted-to-be substrates of P-glycoprotein. In this case, these compounds should be handled with special care if they are to be scaled up to in vivo scenarios. In general, the results obtained by predicting the physicochemical and pharmacokinetic parameters of the compounds indicated that, in most cases, they can be candidates for oral drugs, also highlighting the accessibility of the synthesis, which was consistent with the complexity of the molecule and the synthesis route previously described [29]. However, it should not be ruled out that since they are in silico predictions, they can help to make a more effective follow-up without eliminating the possibility of their use. Therefore, these small, accessible, easily synthesized molecules with antiviral activity could have a promising future because they can be produced on a larger scale for future evaluations.

To identify a possible mechanism of action in vitro, several antiviral strategies have been used. First, we attempted to identify the direct action potential of the compounds on the viral particle, and found that none of them act on the IVP (Figure 1), which is consistent with what was found in a previous study of a group of L-tyrosine phenolic analogous compounds, in which this direct activity on CHIKV was not evidenced [32]. Although the virucidal activity of phenolic compounds against alphaviruses has been evidenced, this has been more often related to the more complex and larger compounds that may have a greater amount of interactions with the particle, such as ginkgolic acid [35], proanthocyanidin [36] or flavonoids, such as quercetagetin [37] and baicalin [38], which additionally have a greater number of hydroxyl groups, which may allow them to form more hydrogen bonds, as well as more than one ring in their structure, which could interact with the aromatic amino acids of the E2 proteins of CHIKV through pi–pi interactions. This is because a greater number of hydroxyl groups are related to a greater capacity for virucidal activity against other arboviruses, such as DENV, as is the case for the flavonoid baicalein, a virucidal compound, compared with its analog, fisetin, which has fewer hydroxyl groups and lacks this activity [39]. This situation could explain in part the absence of activity of the type of compounds in this study.

After the virucidal evaluation, anti-CHIKV activity was observed when the compounds were added prior to infection (pre-treatment). With this strategy, all compounds reduced the amount of viral protein, but only C4, C5 and C6 reduced the viral genome (Figure 2). In the case of these last three compounds, the genomic RNA of CHIKV, being single-stranded positive sense, acts as a messenger RNA, which could indicate that reducing the genome directly leads to a reduction in the translation of the viral protein [10] and that this is one of the inhibition mechanisms of these compounds, which is striking because they are structurally similar compounds, which shows that to inhibit the synthesis of CHIKV viral RNA by this type of compound, the amino group should be methylated (Figure 2 and Figure 8). In the case of C1, C2 and C3, contrary to what has been described with their other three analogs, the reduction of the viral protein is independent of the replication of the viral genome. Furthermore, only these three compounds (C1, C2 and C3) significantly inhibited IVP, and summarizing the previous facts, it could be suggested that the reduction protein that can package the viral RNA and form an infectious viral particle leads to the observed behavior, and that the RNA remains inside the cell (Figure 2C). This confirms the hypothesis of a different mechanism of action, dependent not only on the amino group but also on the halogen, as evidenced by the compounds of Group II.

Therefore, pre-treatment with the first three compounds may be related to a possible prophylactic effect [40], which may be related either to the inhibition of the early stages of infection, which prevents the entry of the viral genetic material into the host–cell, or to the modulation of cellular processes that inhibit the development of the replicative cycle. Therefore, it was proposed to evaluate whether the inhibition of IVP by C1, C2 and C3 was related to the inhibition of adhesion and/or internalization. In Group I (compounds without substitutions in the amino group), only the chlorinated compound inhibited adhesion, unlike the brominated one, which inhibited internalization. In contrast, in Group II, C3 (dimethylated at the amino group) reduced infection in both processes (Figure 3). Among the hypotheses that could be raised about the results of viral adhesion, the antiviral effect could be related either to the binding of the compounds to the cellular receptors or to the A or B domains of the E2 protein, which are the most related to the virus–receptor junction [41]. In addition, the decreased adhesion of both chlorinated compounds could be related to specific interactions with the halogen, perhaps related to the electronegativity of chlorine or its electronic density, which is lower than that of bromine, and could better fit into smaller pockets of the CHIKV E2 protein or cellular receptors [42,43]. In addition, the higher density of bromine also influences the density and electronic distribution in the aromatic ring, resulting in slight deformation, but apparently sufficient to have a different activity than its chlorinated analogs [29,44].

Based on the results of the reduction in the internalization of C2 and C3, bioinformatic methods were used to evaluate whether this inhibition could be related to a possible interaction with the fusion loop of the CHIKV E1 protein. It has been shown that the use of bioinformatics tools for identifying antivirals is useful not only for in silico screening [45] but also for proposing possible mechanisms of action that may be related to the in vitro results obtained [42,45]. This is the case for the antiviral activity of small molecules such as voacangin, which has been shown in other viral models (DENV), despite having less negative binding energies [46]. This in vitro activity could be related to the stability over time of the interaction of the compound with the viral protein [47]. Using tools to evaluate molecular docking, favorable binding energies were obtained for both compounds (slightly lower for C3), and molecular dynamics revealed that the interaction between C3 and the fusion loop was more stable over time (Figure 6) than that between C2 and the fusion loop. Taken together, these results could indicate that the inhibition of internalization induced by C2 could require interactions in addition to those of the fusion loop that could be related to the cellular process of endocytosis. Considering the results of molecular dynamics and molecular docking, it is suggested that the stabilization of complexes between L-tyrosine derivatives and the 3N42 protein is primarily due to interactions mediated by hydrogen bonds and ionic bonds. The degree of relevance of each interaction, as well as the evaluation of the cooperative effects of interactions, should be explored in future studies.

Finally, was evaluated whether the mechanism of action of these compounds is related to processes after the internalization of the virus (post-treatment). Interestingly, none of the compounds in the study inhibited either the protein or the viral genome of CHIKV in this strategy, but the compounds in Groups II and III (C3, C4 and C5, C6, respectively) inhibited the production of IVPs (Figure 4C). This is the first time that the post-treatment antiviral activity of this type of compound has been reported [31,42]. Therefore, it was hypothesized that these compounds could act in stages after replication and translation, such as assembly, maturation and/or release. Knowing that the replication kinetics of the strain used (Appendix A) were consistent with those of other strains previously described [48,49], where it was initially observed, the IVPs that had been internalized and had not yet disassembled, and could be quantified in the monolayer until the eclipse from 1.5 h to 3 h, when the translation and replication processes began to initiate the formation and maturation of PVIs (4.5 h) and its release into the supernatant (6–9 h) until it reached its plateau point (10.5 h). It was confirmed by an addition time test that the effect of these compounds was in the late stages of infection (after the synthesis of protein and the viral genome), since there was no difference in inhibition when these compounds were added at 0 h or at 9 h in any of the cases (Figure 5). Furthermore, the IVPs in the monolayers were also quantified to determine whether the activity was in the release, but no differences were found either in the production of IVPs in the monolayers at the same time as the addition of the compounds, taking into account that it has been evidenced by electron microscopy that CHIKV viral particles could be membrane-bound and could also form on internal membranes [50], which is consistent with the findings of the present study, since there was an increase in IVPs in the monolayer before release in the supernatant (Appendix A). Then, the mechanism could be related to the assembly processes, rather than the release, since no accumulation of IVPs was detected inside the cells, even at the maximum time of the addition of the compounds. Overall, the inhibition of compounds in stages after internalization is independent of protein synthesis and viral genome replication, possibly due to its effect on later stages (such as assembly).

The results of the present study allow us to conclude that although all the Group I compounds showed activity in the early stages of infection (adhesion or internalization), there are differences in the effects they produce, which are caused by halogen. In Group II, the chlorinated compound, C3, showed antiviral activity in both the early and late stages (assembly), while the activity of the brominated compound, C4, differed depending on whether the evaluation was of early or late stages. Moreover, the behavior of Group III compounds (C5 and C6) was the same, regardless of the strategy evaluated.

Considering the in vitro and in silico results, we postulate that of the six evaluated compounds, C3 (a chlorinated tertiary amine) would be an ideal candidate for future drugs, since it inhibits different stages of the CHIKV replicative cycle and is selective for this arbovirus [32]. Additionally, according to the in silico predictions, it would be important to continue with more in-depth investigations that will lead to the elucidation of more specific targets of these compounds in CHIKV and that can be used for future in vivo tests. In conclusion, the findings in these strategies show once again that changes in the type of halogen or the methylation pattern of the amino group can modulate not only the inhibition of infection but also the mechanism by which they do so. Figure 7 summarizes the mechanisms of action proposed for the compounds against CHIKV infection.

## 4. Materials and Methods

### 4.1. Cells and Viruses

The assays were performed on the VERO cell line (ATCC^®^ CCL-81™, derived from kidney epithelial cells of the African green monkey, *Cercopithecus aethiops;* American Type Culture Collection. Manassas, VA, USA). The cells were cultured in DMEM (Dulbecco’s Modified Eagle Medium, Gibco^®^, Grand Island, NY, USA) and supplemented with 2% fetal bovine serum (SFB, Gibco^®^) and 1% antibiotics/antifungals (streptomycin, 10 mg/mL; penicillin, 10,000 U/mL; and amphotericin B, 0.025 mg/mL; Gibco^®^) in a humid atmosphere of 5% CO_2_ at 37 °C. The cells were infected in all cases with the Colombian clinical isolate CHIKV/Col, which belongs to the Asian lineage [25], at a multiplicity of infection (MOI) of 5.

### 4.2. Compounds

For the purposes of this study, the compounds were classified into three groups, according to amino group substitutions (Figure 8). In Group I, the primary amines are compounds (C) C1 and C2; in Group II, the tertiary amines are C3 and C4; and in Group III, the quaternary amines are C5 and C6. In each group, there was a chlorinated compound (C1-C3-C5) and a brominated compound (C2-C4-C6). Chloroquine (CQ) (Sigma Aldrich^®^) was used as a positive control for the inhibition of CHIKV infection at a concentration of 50 µM or ultraviolet (UV) radiation, according to the experimental strategy.

### 4.3. Physicochemical Properties of the Compounds

The prediction of the physicochemical characteristics, “drugability” and absorption, distribution, metabolism and excretion (ADME) properties of the compounds was carried out on the SWISS ADME platform (http://www.swissadme.ch/index.php; accessed on 28 January 2024) of the Swiss Institute of Bioinformatics (https://www.sib.swiss/), as previously described [51]. For this purpose, the SMILES identification of each of the previously modeled compounds was performed on the platform, and the predicted parameters were analyzed as previously described [51,52]. These parameters were topological polar surface area (TPSA), the logarithm of the partition coefficient in n-octanol/water (log P > 5, highly lipophilic; 5 > log P > 2, moderately lipophilic; log P < 2, slightly lipophilic), water solubility (ESOL), gastrointestinal absorption (GIA), passage to the blood–brain barrier (BBB), hydrogen bond acceptor (HBA), hydrogen bond donor (HBD), violations of the Lipinski rules (VLRs) (MW ≤ 500 g/mol, log P ≤ 4.15, ≤10 hydrogen bond acceptors, and ≤5 hydrogen bond donors), structural alerts (PAINS and Brenk) and synthesis accessibility (SA).

### 4.4. In Vitro Treatment with the Compounds

Three antiviral strategies were evaluated to determine the virucidal effect (direct treatment), the effect on cells prior to infection (pre-treatment) and antiviral activity after infection (post-treatment). In all cases, Vero cells were seeded and maintained for 24 h before use. Additionally, for the treatments, the six compounds were used at a concentration of 250 µM, the maximum noncytotoxic concentration at which they have previously demonstrated anti-CHIKV activity [32]. Infections of the CHIKV/Col strain were carried out at a MOI of 5.

For direct treatment, a 1:1 mixture of virus/compound was carried out. These mixtures were preincubated at room temperature for 1 h, after which serial dilutions were made in base 10 for titration by plaque assay on Vero cells [25]. As a positive control, a virus mixture exposed to UV radiation for the same preincubation time was used. In the pre-treatment strategy, the cells were treated with each of the compounds (24 h). Then, the treatment was withdrawn, and the virus was inoculated for 2 h. Afterward, the inoculum was removed, fresh medium was added, and the cells were incubated for 24 h. For post-treatment, the cells were inoculated with the virus for 2 h; then, the inoculum was removed, and each of the compounds was added, followed by incubation for 24 h. In both of the last strategies, the supernatants were collected for the quantification of infectious viral particles (IVPs), and the monolayers were collected for the quantification of the viral genome or fixed for the quantification of viral protein using cell-ELISA as previously described [32]. As a positive control, CQ (50 µM) was used.

#### 4.4.1. Adhesion Inhibition Test

This test was performed to determine whether the compounds interfered with the binding stage of the viral particles to the cell receptors. For this purpose, Vero cells were seeded and incubated for 24 h. The monolayers were treated with the compounds and incubated for 1 h; then, the treatment mixture was withdrawn, and the cells were inoculated with CHIKV/Col and incubated at 4 °C for 1 h to avoid internalization. Finally, the inoculum was removed, and the monolayers were washed twice with PBS 1X, collected and stored at −70 °C until the quantification of IVPs by plaque assay was conducted.

#### 4.4.2. Internalization Inhibition Test

The Vero cells were seeded and incubated for 24 h. The cells were then inoculated with CHIKV/Col and incubated at 4 °C for 1 h to synchronize the infection. Then, treatment with the compounds was performed, and the cultures were incubated for 1 h at 37 °C. The inoculum was removed, and 0.25% trypsin-EDTA was added for 30 s and inactivated. The monolayer was subsequently washed twice with PBS 1X, collected and stored at −70 °C for the subsequent quantification of IVPs by plaque assay.

#### 4.4.3. Addition Time Test

After generating a viral replication curve, the addition time test was performed [53]. After 24 h, VERO cells were seeded and infected with CHIKV/Col. The infection was synchronized for 1 h at 4 °C and then incubated for 2 h at 37 °C. Then, the inoculum was removed, and it was determined as the 0 h experimental point. From this point on, until the first replication cycle was completed (according to the results of the viral replication curve), the compounds were added every 1.5 h. Once the test time was over, the supernatants and monolayers were collected and stored at −70 °C for the subsequent quantification of both intracellular and extracellular IVPs.

#### 4.4.4. Quantification of CHIKV Infection

Infection was quantified using three different methodologies: plaque formation assay, cell-ELISA and qPCR, as previously described [30]. In each case, the percentage of infection was calculated from the CWT (100% infection) according to the units of measurement of each technique.

#### 4.4.5. Statistical Analysis

In each experimental strategy, at least two independent experimental trials were carried out with at least two repetitions (n ≥ 4). Each experiment included control cultures without treatment (CWT) considered as 100.0% of infection. The percentage of infection of each experimental condition was calculated relative to the CWT, according to the units of measurement of each technique used. To evaluate the normality of the data, the Shapiro–Wilk test of normality was used, and, depending on the results, to identify differences between experimental groups and the CWT, Student’s t parametric tests or U Mann–Whitney non-parametric tests were used. Values less than 0.05 (*p* < 0.05) were considered to indicate statistical significance to identify differences between the experimental groups and the CWT.

### 4.5. In Silico Evaluation

#### 4.5.1. Molecular Docking

The in silico interactions between the CHIKV fusion loop present at the envelope protein and the compounds that inhibited virus internalization in vitro were evaluated by molecular docking. The structures of the compounds were modeled with ACD/ChemSketch^®^ 12.01 software (free version). The three-dimensional structure that contained the crystal of the viral envelope proteins (E1 and E2) was selected from the Protein Data Bank database (PDB: 3N42). It was verified that the model had a resolution equal to or less than 3.0 Å, and the quality evaluation of the protein model was carried out using Ramachandran plots and the Structure Assessment module of the Swiss model server (https://swissmodel.expasy.org/; accessed on 23 August 2023). The proteins and ligands used for docking were prepared as previously described [47]. The interaction box (x = −39.036; y = −43.609; z = −30.207; size = 30) with the biologically relevant residues of the fusion loop was defined in the program Autodock tools V1.5.6, according to what is described in the literature [53]. Molecular docking was performed in triplicate with an exhaustiveness value of 10 in the AutoDock Vina program, and the lowest energy positions obtained were selected to evaluate the binding modes as the initial structures of the molecular dynamics simulations [54]. The Protein–Ligand Interaction Profiler server (https://plip-tool.biotec.tu-dresden.de/plip-web/plip/index; accessed on 23 August 2023) was used to examine interactions between the anti-internalization compounds and the viral protein [55].

#### 4.5.2. Molecular Dynamics Simulations

Molecular dynamics (MD) was performed as previously described [52]. The protein–ligand complex structures for each molecular dynamics (MD) simulation were established based on the ideal posture score obtained from the molecular docking analysis. SwissParam was utilized to generate topology files for the ligand structures, facilitating the execution of the MD simulations [56]. For the MD simulations, the CHARMM27 force field was used in the Gromacs 2021.2 package [57]. The protein–ligand complexes were initially prepared by conducting energy minimization in water. A TIP3P water molecule model was employed to center each system within a cubic box using specific vectors, with the steepest descent energy value utilized for this process [58]. Next, for energy minimization, an MD equilibration phase was conducted, beginning with a 2 ns simulation using an isochoric-isothermal ensemble (NVT), followed by a 2 ns simulation using an isothermal-isobaric ensemble (NPT). To neutralize each protein–ligand complex, five Na+ counterions were introduced into the continuous solvent phase [59]. The system was deemed to have achieved proper energy minimization when it remained within a tolerance of 10 kJ/mol. To maintain bonds subject to holonomic constraints, the LINCS method was employed [60]. The modified Berendsen coupling V-rescale technique in the NVT ensemble was used to control the temperature of the complexes [61]. The Parrinello–Rahman coupling algorithm was employed to establish the NPT ensemble for pressure control at 1 atm. Each protein–ligand complex underwent a 100 ns MD simulation under periodic boundary conditions at 310 K and 1 atm, with a short-range van der Waals cutoff of 1.2 nm. Subsequent assessments, including root mean square deviation (RMSD), root mean square fluctuation (RMSF) and hydrogen bond analysis, were conducted using the Gromacs software suite and visual molecular dynamics (VMD) v1.9.4 [62]. RMSD values between 0.2 and 0.6 can indicate low fluctuations in the complex topology [63]. In our case, with RMSD values close to 0.4 or lower, we can suggest the stability of the complexes.

## Figures and Tables

**Figure 1 ijms-25-07972-f001:**
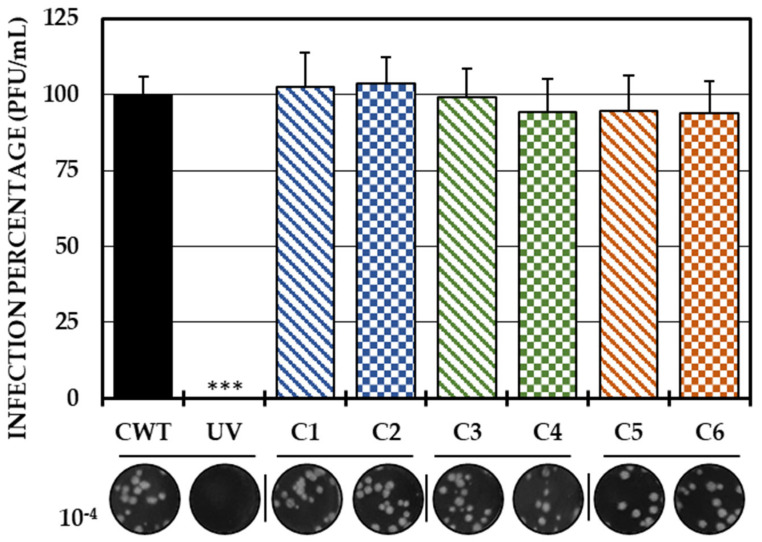
Direct effect of compounds on IVP. Percentages of infection calculated according to the results obtained by the plaque assay (plaque forming units per milliliter, PFU/mL) of the mixtures between the virus and each of the compounds. The inhibition control of the technique was UV radiation (0.0%, <1.0 × 10^2^ PFU/mL). At the bottom is shown the representative plaques of titration on VERO cells corresponding to each experimental condition and the log title of the images. In all cases, control cultures without treatment (CWTs) were assumed as 100.0% infection. The asterisks indicate statistically significant differences with respect to the control without treatment (*** *p* < 0.001; Student’s *t* test), and the error bars indicate the standard error of the mean (n = 6).

**Figure 2 ijms-25-07972-f002:**
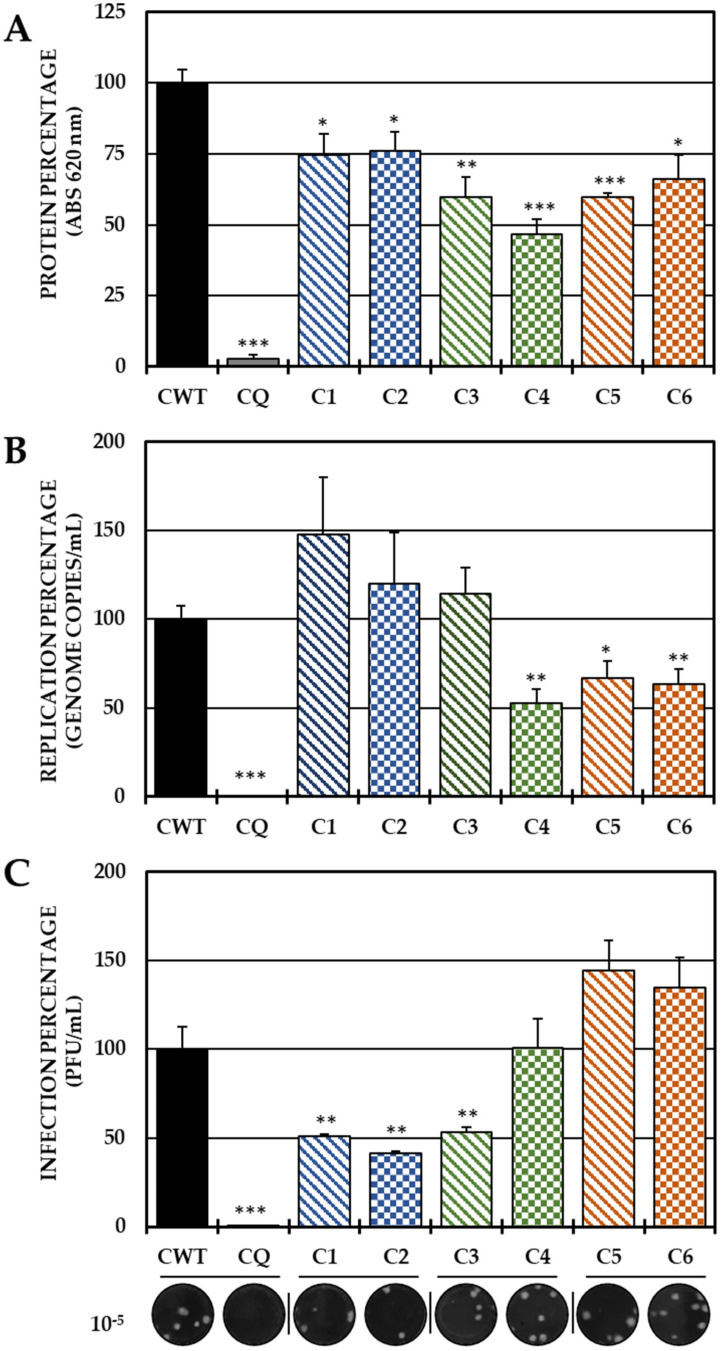
Effect of tyrosine-derived compounds on pre-treatment CHIKV infection. The relative percentages of infection were calculated according to the results obtained using the cell-ELISA (**A**) and qPCR (**B**) assays in monolayers and through the plaque assay (PFU/mL) of the supernatants (**C**) of the pre-treatment antiviral assay (treatment prior to CHIKV/Col virus inoculation at a multiplicity of infection (MOI) of 5). The inhibition control technique was 50 µM CQ (2.9% infection in the case of cell-ELISA (A); 0.5% infection and 8.6 × 10^2^ genomic copies/mL in the case of qPCR (**B**) and 0.2% infection and 1.9 × 10^4^ PFU/mL in the case of the plaque assay (**C**)). At the bottom is shown representative plaques of titration on VERO cells corresponding to each experimental condition and the log title of the images. In all cases, the CWT was assumed to be 100.0% infection (* *p* < 0.05, ** *p* < 0.01 and *** *p* < 0.001; Student’s *t* test), and the error bars indicate the standard error of the mean (n = 4).

**Figure 3 ijms-25-07972-f003:**
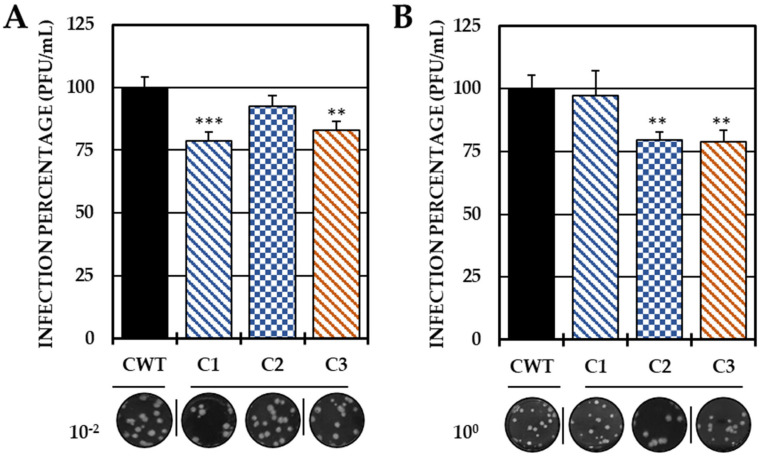
Effect of compounds in the early stages of infection. Percentages of infection calculated according to the results obtained by the plaque assay (PFU/mL) of the supernatants collected from the adhesion inhibition assay (**A**) and from the internalization inhibition assay monolayers (**B**). In all cases, the CWT was assumed to be 100.0% infection. At the bottom is shown representative plaques of titration on VERO cells corresponding to each experimental condition and the log title of the images. In all the experimental conditions, the cells were infected with CHIKV/Col at an MOI of 5. The asterisks indicate that the results were significantly different from those in the CWT group (** *p* < 0.01 and *** *p* < 0.001; *t* test). Two independent experiments were carried out with three replicates each (n = 6). The error bars indicate the standard error of the mean.

**Figure 4 ijms-25-07972-f004:**
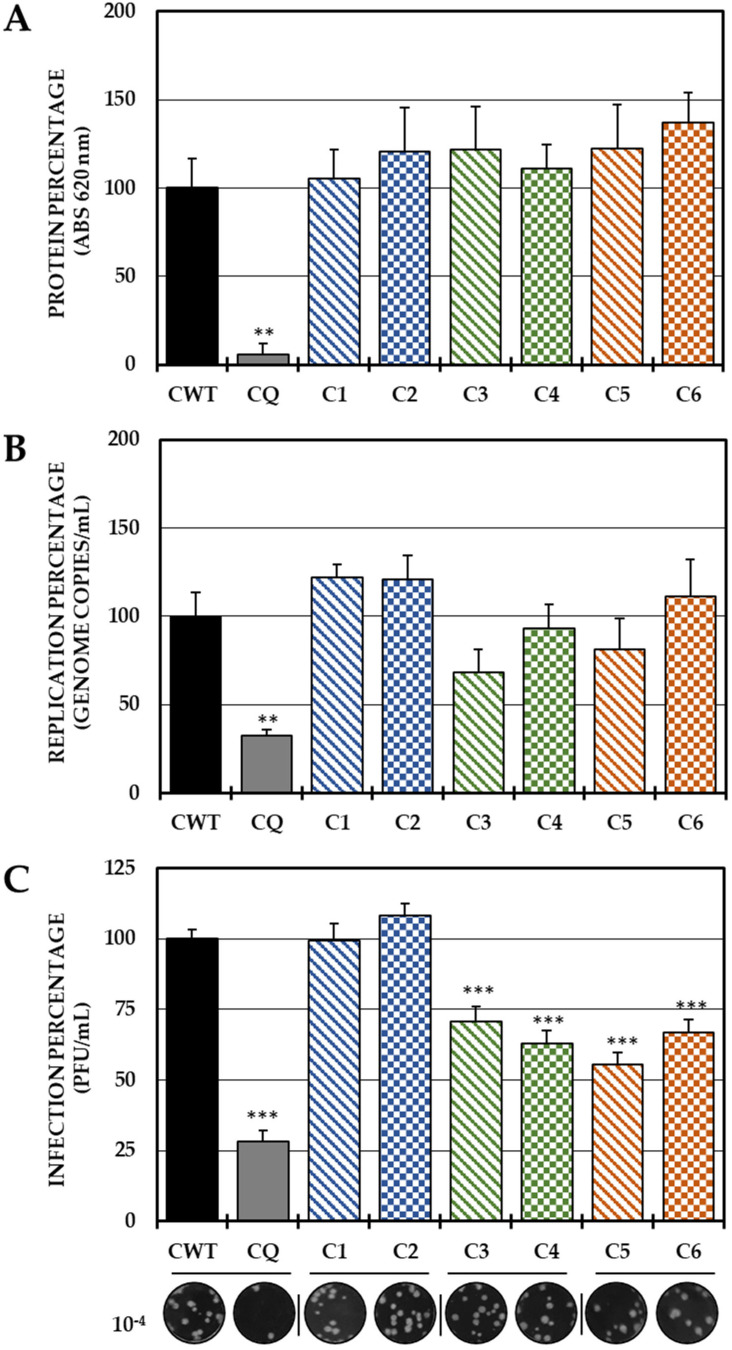
Effect of dH-Y on post-treatment CHIKV infection. The relative percentages of infection were calculated according to the results obtained by the cell-ELISA (**A**) and qPCR (**B**) assays in monolayers and by the plaque assay (PFU/mL) of the supernatants of the post-treatment antiviral assay (treatment after CHIKV/Col virus inoculation at an MOI of 5). The inhibition control technique was 50 µM chloroquine (2.9% infection in the case of cell-ELISA (**A**), 0.5% infection, 8.6 × 10^2^ genomic copies/mL in the case of qPCR (**B**) and 28.3% infection, 8.1 × 10^5^ PFU/mL (**C**)). At the bottom is shown representative plaques of titration on VERO cells corresponding to each experimental condition and the log title of the images. In all cases, the CWT was assumed as 100.0% infection (** *p* < 0.01 and *** *p* < 0.001; Student’s *t* test), and the error bars indicate the standard error of the mean (n = 4).

**Figure 5 ijms-25-07972-f005:**
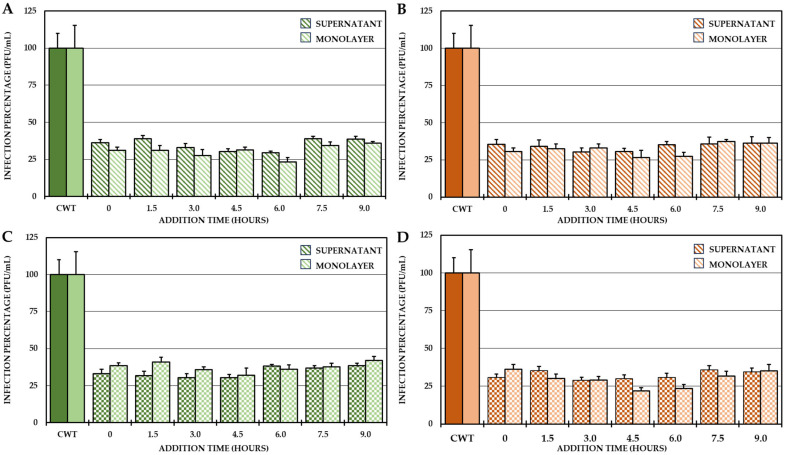
Effect of compounds on the production of IVP in the CHIKV/Col replication cycle. Percentages of infection calculated according to the results obtained by plaque assay (PFU/mL) in both the supernatant and monolayer of VERO cells infected with the CHIKV/Col virus at an MOI of 5 from the addition time assay for the compounds of Group II (**A**,**C**) and Group III (**B**,**D**). All the experimental samples were collected at 10.5 h after the removal of the inoculum. In all cases, the CWT was assumed to be 100.0% infection. All experimental conditions were significantly different from those of the control without treatment (*p* < 0.05; Student’s *t* test), and the error bars indicate the standard error of the mean (n = 4).

**Figure 6 ijms-25-07972-f006:**
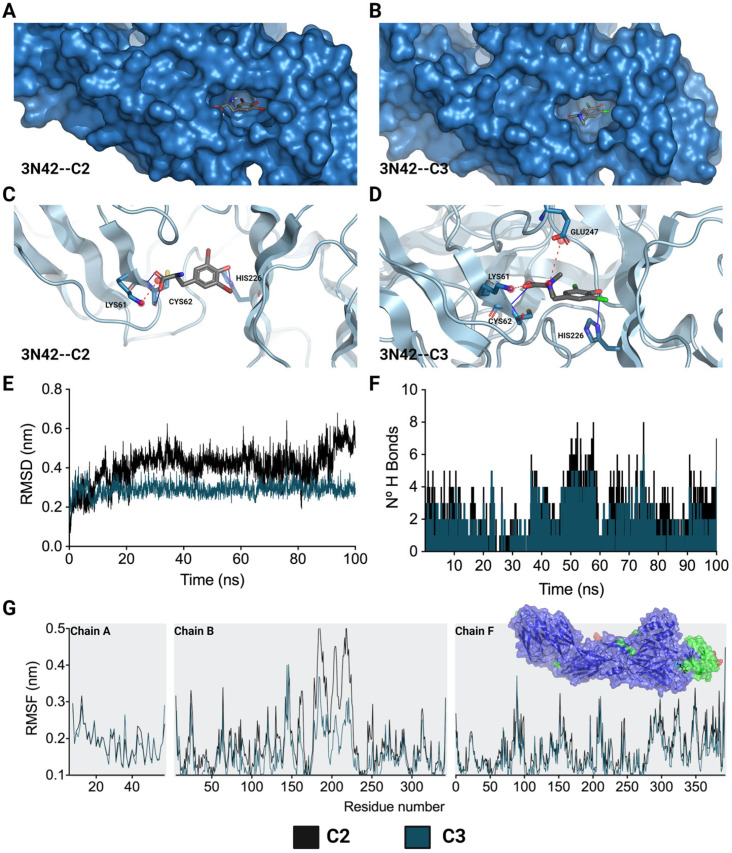
Lowest-energy docked pose and stability of complexes formed by L-tyrosine derivatives with 3N42. C2-3N42 binding pocket and interaction modes (**A**,**C**). C3-3N42 binding pocket and interaction modes (**B**,**D**). Root mean square deviation (RMSD) (**E**). Hydrogen bonds (**F**). Root mean square fluctuation (RMSF) (**G**). In (**C**,**D**), the interactions are represented as hydrogen bonds (blue lines) and ionic bonds (red dotted lines). In (**E**–**G**), the black and dark green lines correspond to C2 and C3, respectively.

**Figure 7 ijms-25-07972-f007:**
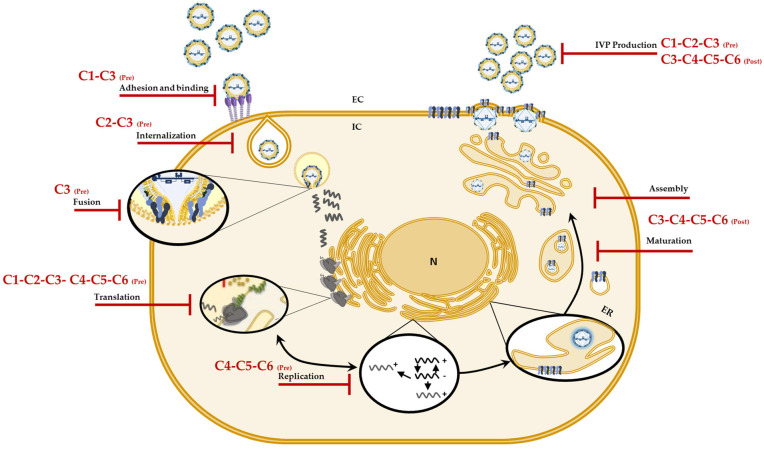
Possible mechanisms of action of the compounds against CHIKV. Graphic representation of the sites on the CHIKV replicative cycle on which, according to the results, the compounds (C1 to C6) could have an effect and jointly explain the in vitro antiviral activity.

**Figure 8 ijms-25-07972-f008:**
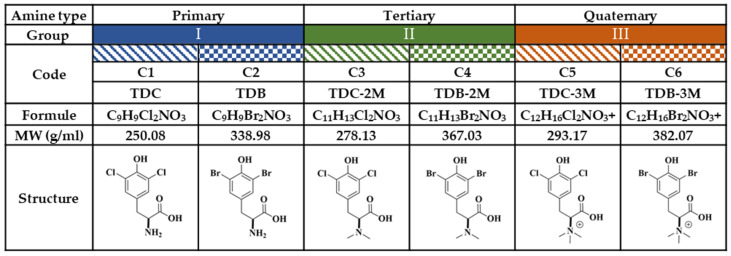
Study compounds. The six dH-Y were classified into three groups according to the substitution of the amino group. Each group has a chlorinated compound (Group I, 2-amino-3-(3,5-dichloro-4-hydroxyphenyl) propanoic acid; Group II, 3-(3,5-dichloro-4-hydroxyphenyl)-2-(dimethylamino) propanoic acid; and Group III, 1-carboxy-2-(3,5-dichloro-4-hydroxyphenyl)-N,N-trimethylethan-1-aminium; corresponding to C1, C3 and C5, respectively); and a brominated compound (Group I, 2-amino-3-(3,5-dibromo-4-hydroxyphenyl) propanoic acid; Group II, 3-(5-dibromo-4-hydroxyphenyl)-2- (dimethylamino) propanoic acid; and Group III, 1-carboxy-2-(3,5-dibromo-4-hydroxyphenyl)-N, N,N-trimethylethan-1-aminium; corresponding to C2, C4 and C6, respectively).

**Table 1 ijms-25-07972-t001:** Physicochemical and pharmacokinetic parameters of dH-Y.

		TPSA	HBA	HBD	#Rot	Log P	ESOL		Medicinal Chemistry				Pharmacokinetics			
		(Å^2^)					Log S	Class	SA	LV	PAINS	Brenk	GIA	BBB	CYPi	P-gps
**I**	**C1**	83.55	4	3	3	0.74	−1.17	V-S	1.96	0	0	0	High	No	No	No
	**C2**	83.55	4	3	3	0.91	−1.80	V-S	2.13	0	0	0	High	No	No	No
**II**	**C3**	60.77	4	2	4	1.84	−1.86	V-S	2.15	0	0	0	High	Yes	No	No
	**C4**	60.77	4	2	4	1.96	−2.49	S	2.31	0	0	0	High	Yes	No	No
**III**	**C5**	57.53	3	2	4	0.37	−3.39	S	2.29	0	0	1	High	Yes	No	Yes
	**C6**	57.53	3	2	4	0.58	−4.01	Md-S	2.44	0	0	1	High	Yes	No	Yes

The treatment groups, compounds and each of the predicted characteristics are highlighted in bold. TPSA: topological polar surface area; HBA: number of H-bond acceptors; HBD: number of H-bond donors; #Rot: number of rotatable bonds; Log P: consensus of the partition coefficient between n-octanol and water (o/w); ESOL: estimated aqueous solubility; Log S: decimal logarithm of molar solubility in water; SA: synthetic accessibility; LV: Lipinski violations; PAINS: panassay interference compounds; Brenk: Brenk’s structural alert; GIA: gastrointestinal absorption; BBB: blood–brain barrier permeation; CYPi: cytochrome P450 inhibitor; P-gps: glycoprotein P substrate.

## Data Availability

The datasets used and analyzed during the current study are available from the corresponding author on reasonable request.

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
