# Peer review of "The Mechanism of Action of L-Tyrosine Derivatives against Chikungunya Virus Infection In Vitro Depends on Structural Changes"

_ijms, 2024, doi:10.3390/ijms25147972_

Round 1

Reviewer 1 Report

Comments and Suggestions for Authors

The authors investigated the mechanism underlying the inhibition of chikungunya virus (CHIKV) replication by six L-tyrosine derivatives. Both in vitro and in silico approaches were used and the results were convincing, leading the authors to propose a working model that deciphers how these chemicals perform anti-CHIKV activities. Overall, I would recommend this manuscript to be published, if these questions/suggestions could be addressed:

Line 25 mentions C1, C2, C3 inhibited IVP while line 28 states C3, C4, C4 and C6 inhibited IVP. Please double check and correct.

Introduction could be improved by deleting some background irrelevant to this study (epidemics), but adding a summary of the findings of this work.

The infection rate is always perfectly 100% without any treatment, but sometimes the treatment result shows an increase of infection rate that is higher than 100%, especially for C1 and C2 in multiple experiments. Is it possible to optimize the statistics?

The last paragraph of page 9 requires a clearer explanation on the molecular dynamics simulation. What RMSD is it? How to define a system being stable in the experiment?

Figure 6G is confusing. How many chains does the viral protein have? Why only the RMSF results of chains A, B and F are shown? Does it only have three chains? It is stated that the active site shows the highest RMSF, can the authors emphasize the location of the site in the model?

Author Response

RESPONSE TO REVIEWER COMMENTS #1

The authors investigated the mechanism underlying the inhibition of chikungunya virus (CHIKV) replication by six L-tyrosine derivatives. Both in vitro and in silico approaches were used and the results were convincing, leading the authors to propose a working model that deciphers how these chemicals perform anti-CHIKV activities. Overall, I would recommend this manuscript to be published, if these questions/suggestions could be addressed:

Comment 1-1

Line 25 mentions C1, C2, C3 inhibited IVP while line 28 states C3, C4, C4 and C6 inhibited IVP. Please double check and correct.

Answer from author:

We checked and improved this statement: Only C3, C4, C5 and C6 inhibited IVP when the cells were exposed after infection. Line 28.

Comment 1-2

Introduction could be improved by deleting some background irrelevant to this study (epidemics), but adding a summary of the findings of this work.

Answer from author:

We appreciate the reviewer's suggestion and have removed some parts of the introduction related to epidemiology. However, we consider it inappropriate to include in that section the summary of the main findings because we already described them in the abstract, and we preferred to end the introduction with the objective of the study to respond to it in the next section.

Comment 1-3

The infection rate is always perfectly 100% without any treatment, but sometimes the treatment result shows an increase of infection rate that is higher than 100%, especially for C1 and C2 in multiple experiments. Is it possible to optimize the statistics?

Answer from author:

Because the results are graphing in percentage of infection relative to the control without treatment (CWT), in some cases the bar can be results both below as above 100%. However, it is important to clarify that the statistical analysis was performed with the original measurement units: plaque formation units (plaque formation assay), absorbance (cell-ELISA) and genome copies (qPCR). In the specific case mentioned by the reviewer (C1 and C2), we again checked the raw data, but we did not find outliers or significant differences with respect to the CWT.

Comment 1-4

The last paragraph of page 9 requires a clearer explanation on the molecular dynamics simulation. What RMSD is it?

Answer from author:

The RMSD is already described in Line 249. However, we improved the description (lines 250-251).

Comment 1-5

How to define a system being stable in the experiment?

Answer from author:

Although there are currently no established guidelines regarding appropriate RMSD values to determine the stability of a complex, numerous articles have suggested that RMSD values between 0.2 and 0.6 can indicate low fluctuations in the complex topology. In our case, with RMSD values close to 0.4, we can suggest the two complexes are stable. Here are some examples: doi: 10.1080/07391102.2018.1441069; doi: 10.1016/j.foodchem.2020.128933; doi: 10.3390/ijms23031771. To clarify this point, we included in the Materials and Methods section a description of the stability parameters used in our study (lines 577-579).

Comment 1-6

Figure 6G is confusing. How many chains does the viral protein have? Why only the RMSF results of chains A, B and F are shown? Does it only have three chains? It is stated that the active site shows the highest RMSF, can the authors emphasize the location of the site in the model?

Answer from author:

As the reviewer correctly described, the 3N42 protein downloaded from the PDB server has only three chains (https://www.rcsb.org/structure/3n42), which are described in the crystallographic description as chains A, B, and F (Figure 6). For this reason, the present investigation retained the notation and number of chains described in the crystallographic structure. For better understanding, we improved Figure 6 (Line 268). The RMSF results are described in lines 255 to 263, and the active site is described in line 258.

Reviewer 2 Report

Comments and Suggestions for Authors

The manuscript describes the investigation of the antiviral action mechanisms of L-tyrosine derivatives against Chikungunya Virus infection in vitro, which has been found to be related to structural changes.

The topic fits the special issue “Molecular Research on Antiviral Mechanism,” the scope is adequate and the methods are appropriate to acquire experimental results complemented by in silico results supporting the conclusions.

On the methodology, the experimental investigation has adequate details for reproducibility. However, the in silico part has compromised reproducibility due to missing a comprehensive supplementary file with detailed starting and final geometry, molecular structures and energies of MD simulations. Perhaps some of the docking results not presented in the manuscript should also go inside this DI. No results should be not reported. The overall Section 4 excessively refers to previous studies, which dilutes the novelty of the study. Although Student’s t test has been mentioned in the figure captions, it has not been introduced in Section 4.

The full terms of each abbreviation should be introduced at first appearance.

What are the images at the bottom of Fig. 1-4 with “10-n”?  

Double check Line 228 value 6.10 ± 0.00 kcal/mol.

It is hard to follow which is C2 and C3 in Fig. 6, please mark each panel, and add color indication which color is which in E, F, G. Additionally, these last 3 panels have not been discussed. What does N° refer to?

Fig. 5, the candidate compounds’ performance is far inferior to the control. This should be discussed, as well as the practical implications.

Last but not least, the authors have identified the pronounced structure-activity and structure-specificity relationship in the tyrosine peptide. In particular, its sidechain and the wide potential for functional substitution on the aromatic ring and its role in intra-molecular interactions that stabilise specific structures and give rise to activity. Something that can be synthetically modulated to give rise to activity, such as the anti-viral nature of halogenated tyrosyl residues, as detailed in the current work. They should highlight this atomistic-scale function and specificity by citing work that has focused on this. Perhaps to evolve the relevant text as follows: (Page-2, Lines 87-91)

"This allows not only larger amounts of compounds to be obtained but also different functional groups of L-tyrosine to be substituted, generating structural changes (halogens in the ring, phenolic hydroxyl groups, esterification of the carboxyl group or methylation of the amino group)[32]; with structure and  function-specific intramolecular interactions in tyrosyl-based peptides being quantitatively characterised.[New-ref] These site-specific synthetic substitutions giving rise to the high-efficacy synthetic analogues have shown activity against HIV-1 [33], ZIKV [34] and CHIKV [34]."

[New-Ref] GA Chass, et. al., The role of enhanced aromatic-electron donating aptitude of the tyrosyl sidechain with respect to that of phenylalanyl in intramolecular interactions, Eur. Phys. J. D 20, 481-497 (2002)

With refs 33-70 becoming 34-71 and relevant citations in text being indexed by +1.

Author Response

RESPONSE TO REVIEWER COMMENTS #2

The manuscript describes the investigation of the antiviral action mechanisms of L-tyrosine derivatives against Chikungunya Virus infection in vitro, which has been found to be related to structural changes. The topic fits the special issue “Molecular Research on Antiviral Mechanism,” the scope is adequate and the methods are appropriate to acquire experimental results complemented by in silico results supporting the conclusions.

Comment 2-1

On the methodology, the experimental investigation has adequate details for reproducibility. However, the in silico part has compromised reproducibility due to missing a comprehensive supplementary file with detailed starting and final geometry, molecular structures and energies of MD simulations.

Answer from author:

According to the reviewer's recommendations, the following information has been included in the Results section: “In an initial assessment, the topology of the complexes was verified for C2 and C3 (Supplementary Figure 3A-B, respectively) at 0 ns (in cyan) and 100 ns (in red) time points. Both compounds remained within the protein pocket at the active site of 3N42” (lines 246–249 and lines 582–583). The system stability was evaluated by means of the RMSD (root mean square deviation) value for the protein backbone C-α in complex with L-tyrosine derivatives” (Lines 249-251). Additionally, a supplementary image with the initial and final topologies was generated (Supplementary Figure 3A-B).

Comment 2-2

Perhaps some of the docking results not presented in the manuscript should also go inside this DI.  No results should be not reported.

Answer from author:

We apologize to the evaluator, but we do not identify where he requests that we should show the docking information, since it is all described in the text (lines 232 to 234) and shown in Figure 6 (A to D). However, to improve the docking information, the coordinates of the interaction box were included in the methodology (lines 544 – 545).

Comment 2-3

The overall Section 4 excessively refers to previous studies, which dilutes the novelty of the study.

Answer from author:

According to the recommendation, we checked Section 4 “Materials and Methods”, and we deleted some references from previous studies (especially those related to in vitro studies). However, references related to in silico assays must remain because they are free software and because copyright is mandatory for reference.

Comment 2-4

Although Student’s t test has been mentioned in the figure captions, it has not been introduced in Section 4.

Answer from author:

We appreciate the comment. The description of the Student’s t test was added to subsection 4.4.5. (Statistical analysis) (Lines 529-531).

Comment 2-5

The full terms of each abbreviation should be introduced at first appearance.

Answer from author:

We would like to thank the reviewer for identifying this error. Due to a change in the MDPI Journals template, most of the abbreviations were found in the methodology at the end of the writing and not at the beginning. For this reason, we described all of them in their first appearance in the article. Lines 25-26, 41, 69, 81-82, 90, 98-99, 102, 113-114, 119, 144, 148, 167, 244, and 255.

Comment 2-6

What are the images at the bottom of Fig. 1-4 with “10-n”?

Answer from author:

We appreciate this comment because we were unable to clarify the explanation at the bottom of several figures. For this reason, we made the appropriate changes to the footnotes of Figure 1 (lines 146-147), Figure 2 (lines 170-172), Figure 3 (lines 185-186), and Figure 4 (lines 200-202).

Comment 2-7

Double check Line 228 value 6.10 ± 0.00 kcal/mol.

Answer from author:

According to the edition of the journal, the “negative symbol” was ubicated in the above line (233). We hope that this mistake has been corrected in the final revised version of the manuscript before its publication.

Comment 2-8

It is hard to follow which is C2 and C3 in Fig. 6, please mark each panel, and add color indication which color is which in E, F, G. Additionally, these last 3 panels have not been discussed. What does N° refer to?

Answer from author:

Following the reviewer's recommendation, we included descriptions of the complexes formed in figures A, B, C, and D. This allows the reader to easily identify the compounds forming each complex. Similarly, color legends for E, F, and G have been added to the images (Figure 6). The abbreviation "N°" refers to the number of hydrogen bonds formed throughout the dynamics. A description of figures 6 E, F, and G can be found in lines 249-267, and the stability of the formed complexes is discussed in lines 368-372.

Comment 2-9

Fig. 5, the candidate compounds’ performance is far inferior to the control. This should be discussed, as well as the practical implications.

Answer from author:

We are not sure of the reviewer’s comment. However, we would like to clarify that Figure 5 does not have an inhibition control (such as chloroquine). Moreover, the implications of infection reduction at each treatment time point have already been discussed (Lines 393-407).

Comment 2-10

Last but not least, the authors have identified the pronounced structure-activity and structure-specificity relationship in the tyrosine peptide. In particular, its sidechain and the wide potential for functional substitution on the aromatic ring and its role in intra-molecular interactions that stabilise specific structures and give rise to activity. Something that can be synthetically modulated to give rise to activity, such as the anti-viral nature of halogenated tyrosyl residues, as detailed in the current work. They should highlight this atomistic-scale function and specificity by citing work that has focused on this. Perhaps to evolve the relevant text as follows: (Page-2, Lines 87-91)

Answer from author:

We appreciate the reviewer’s suggestion, but in this work, we did not evaluate peptides to which tyrosyl substitutions were made but rather compounds derived from tyrosine. This is why the phrase suggested by the reviewer (“with structure and function-specific intramolecular interactions in tyrosyl-based peptides being quantitatively characterized. [New-ref]”) it is not related to the described in that paragraph since it refers to substitutes of the chemical nucleus Tyrosine. However, we included the sentence “These substitutions, in addition to the known aromatic π-electron donating ability of tyrosine [30], could generate changes and improvements in both activity and mechanism”, which is consistent with the reference proposed by the reviewer, but in another context, they are better related (Lines 87-89).

Round 2

Reviewer 2 Report

Comments and Suggestions for Authors

All the issues have been addressed.